# High-Fat Diet Enhances the Liver Metastasis Potential of Colorectal Cancer through Microbiota Dysbiosis

**DOI:** 10.3390/cancers14112573

**Published:** 2022-05-24

**Authors:** Yina Yu, Yangke Cai, Bin Yang, Siyuan Xie, Wenjuan Shen, Yaoyi Wu, Ziqi Sui, Jianting Cai, Chao Ni, Jun Ye

**Affiliations:** 1Department of Gastroenterology, The Second Affiliated Hospital, Zhejiang University School of Medicine, Hangzhou 310009, China; 21918296@zju.edu.cn (Y.Y.); 22018114@zju.edu.cn (Y.C.); 21818202@zju.edu.cn (B.Y.); 3140102064@zju.edu.cn (S.X.); jhf@zjhu.edu.cn (W.S.); wuyaoyi@zju.edu.cn (Y.W.); jtcai6757@zju.edu.cn (J.C.); 2Department of Gastroenterology, The First People’s Hospital of Linping District, Hangzhou 310009, China; wenqisui@jmsu.edu.cn; 3Department of Breast Surgery, The Second Affiliated Hospital, Zhejiang University School of Medicine, Hangzhou 310009, China

**Keywords:** colorectal cancer, liver metastasis, high-fat diet, microbiota dysbiosis, *Desulfovibrio*, tumour microenvironment

## Abstract

**Simple Summary:**

High-fat diet (HFD) is hypothesized to induce gut dysbiosis and promote colorectal cancer (CRC). However, the specific mechanisms involved require investigation. In this study, we established an animal model and utilized 16S sequencing to determine the effects of HFD on gut microbiota, as well as on the colon and liver. Furthermore, due to the abundance of *Desulfovibrio* (DSV) in the faecal samples of HFD-fed rats and CRC hepatic metastasis patients, we also conducted a DSV gavage animal experiment to determine the role of DSV in CRC development. Our study confirmed that HFD could cause microbiota dysbiosis, especially DSV enrichment, and may promote CRC initiation and metastasis.

**Abstract:**

Obesity, metabolic changes, and intestinal microbiota disruption significantly affect tumorigenesis and metastasis in colorectal cancer (CRC). However, the relationships among these factors remain poorly understood. In this study, we found that a high-fat diet (HFD) promoted gut barrier dysfunction and inflammation in the colorectum and liver. We further investigated gut microbiota changes through 16S rRNA sequencing of faecal samples from HFD-fed rats and CRC hepatic metastasis patients and found an abundance of *Desulfovibrio* (DSV). DSV could also induce barrier dysfunction in the colorectum and inflammation in the colorectum and liver, suggesting that it contributes to the formation of a microenvironment conducive to CRC tumorigenesis and metastasis. These findings highlight that HFD-induced microbiota dysbiosis, especially DSV abundance, could promote CRC initiation and metastasis.

## 1. Introduction

According to GLOBOCAN 2018, colorectal cancer (CRC) is the third most common cancer and the second most significant cause of cancer-related deaths [1]. The liver is the most common site for CRC metastasis [2]. There is convincing evidence that being overweight is a risk factor for several malignancies, including breast, colorectal, prostate, and pancreatic cancers [3,4]. Serum and tissue lipid levels are abnormally altered in CRC [5]. Metabolic changes induced by obesity have a great influence on the microenvironment, which is beneficial for tumour initiation and progression [6]. In a previous study, low-density lipoprotein cholesterol (LDL-c) levels correlated positively with liver metastasis in CRC [7].

Studies have suggested that a high-fat diet (HFD) results in gut microbial alterations [8,9]. HFD-related gut bacterial changes may influence the secretory, absorptive, and immune functions of the gut [10]. HFD can regulate the gut microbial composition and intestinal inflammation, to increase the susceptibility of the intestine to carcinogenic factors and exacerbate CRC [11,12]. The faecal microbiota of CRC patients could increase the numbers of polyps, and promote intestinal dysplasia and proliferation, and inflammatory markers, in germ-free and conventional mice [13]; this indicates that changes in intestinal microbiota may induce gut inflammation and carcinogenesis. As the main product of gut gram-negative microbiota, lipopolysaccharides (LPS) promote intestinal inflammation and CRC progression [14]. An inflammatory microenvironment shaped by gut microbiota can promote tumorigenesis in the colorectum [15].

Disruption of tight junctions in the gut may cause the translocation of microbiota and microbial products [16]. Intestinal barrier dysfunction can increase the portal influx of pathogen-associated molecules into the liver through the gut-liver axis [17]. Paget’s “seed and soil” hypothesis (1889) posits that the microenvironment of specific metastasis organs are suitable for tumour colonization and outgrowth [18]. Research has shown that changes in the liver microenvironment influence the metastasis potential of CRC [19,20]. LPS can activate PI3K/AKT signalling and increase the adhesiveness and metastatic capacity of CRC cells [21]. The formation of a pre-metastatic niche in the liver is induced by inflammation [22]. Besides intestinal microbiota disorders, HFD can also induce liver inflammation through dysbiosis and promote metastasis [23].

Intestinal microbiota disruption is associated with metabolic diseases and cancers, including breast, liver, and colorectal cancers [24,25,26]. Especially in CRC, there is an abundance of flora, including Bacteroides fragilis, Escherichia coli, Streptococcus gallolyticus, Enterococcus faecalis, and Fusobacterium nucleatum [15,27]. Short-chain fatty acids (SCFAs) are widely investigated metabolites of intestinal microbiota, which can protect intestinal mucosa from inflammation [24]. As an SCFA, butyrate inhibits CRC tumour progression through multiple mechanisms, including anti-inflammatory and immunomodulatory effects [28]. However, microbiota metabolites are not always beneficial. H_2_S can stimulate angiogenesis and support tumour proliferation [29]. A prospective study of intestinal microbiota diversification demonstrated that sulphate-reducing bacteria (SRB), such as *Desulfovibrio* (DSV), might contribute to CRC development [30]. DSV levels in the saliva of CRC patients were elevated compared to healthy controls [31]. Furthermore, relative enrichment of DSV was reported in an analysis of multiomics data from a large cohort of CRC patients [32].

However, the carcinogenesis and metastasis induced by DSV in CRC remain poorly understood. The mechanism by which HFD-related flora disruption influences CRC initiation and metastasis is also unclear. In this study, we aimed to determine changes in intestinal inflammation and barrier function in rats fed an HFD. To investigate the microbiota dysbiosis induced by HFD, and the correlations among flora composition, CRC, and CRC liver metastasis, we sequenced the 16S bacterial gene from faecal samples obtained from HFD mice and CRC liver metastasis patients. Through analysis of sequencing data, we further evaluated the influence of DSV on carcinogenesis and the liver metastasis potential of CRC.

## 2. Materials and Methods

### 2.1. Animals and Treatment

Adult male Wistar rats and adult male BALB/c mice were purchased from Shanghai SLAC Laboratory Animal (Shanghai, China). Newly purchased six-week-old murine animals were housed for one week to normalize the gut microbiota and were randomised to control (standard diet, n = 7), HFD (high-fat diet, n = 7), and HFD+antibiotics (high-fat diet and antibiotics, n = 8) groups. Each group was fed their respective diet for eight weeks. The HFD provided 60%, 20%, and 20% calories as fat, carbohydrates, and proteins, respectively. Rats were housed in a temperature-controlled environment (23 ± 1 °C with an average humidity of 60 ± 1% and 12-h light/dark cycle) with free access to food and water. To deplete the gut microbiota, an antibiotic cocktail containing ampicillin (0.2 g/L; FUSHENBIO, Shanghai, China), neomycin (0.2 g/L; FUSHENBIO), metronidazole (0.2 g/L; FUSHENBIO), and vancomycin (0.1 g/L; FUSHENBIO) was added to the water in the HFD+antibiotics group for two weeks (every other 2 weeks) until the end of experiment [33]. The body weights of the animals were measured weekly. The mice were randomly assigned to the control (n = 7), DSV (n = 7), and DSV+antibiotics (n = 8) groups and fed a standard diet. To deplete gut microbiota, an antibiotic cocktail, containing ampicillin (0.2 g/L; FUSHENBIO), neomycin (0.2 g/L; FUSHENBIO), metronidazole (0.2 g/L; FUSHENBIO), and vancomycin (0.1 g/L; FUSHENBIO) was added to the water and fed to the mice twice a day for two weeks by gavage [33,34]. After two weeks, the mice in the DSV and DSV+antibiotics groups were fed with DSV by gavage (approximately 1.4 × 10^9^ CFU/mL, 10 μL/g) for eight weeks. The bodyweight of the animals was measured weekly. At the end of the feeding period, the mice and rats were anesthetized by intraperitoneal injections of sodium pentobarbital (60–75 and 45 mg/kg body weight, respectively). The experimental protocols were approved by the Laboratory animal management and ethics committee of Zhejiang Chinese Medical University (Hangzhou, China).

### 2.2. Faecal Sample Collection

A total of 23 CRC patients diagnosed between January 2019 and January 2020 were recruited from The Second Affiliated Hospital of Zhejiang University (Hangzhou, China). Among the patients, 13 were diagnosed as CRC with liver metastasis and 10 as CRC without metastasis. We collected stool samples and immediately stored them at −80 °C in a refrigerator for analysis. The exclusion criteria were diabetes, infectious diseases, diarrhoea or constipation, multiple intestinal adenomas, special dietary habits, and antibiotic use during the past month. The Ethics Committee of The Second Affiliated Hospital of Zhejiang University School of Medicine approved the study (No. I2021001859). Fresh faeces were obtained from the rats before they were sacrificed by massaging the abdomen. Then, the faeces were promptly placed into liquid nitrogen and stored at −80 °C. The human and rat faecal samples were submitted to Sangon Biotech Co., Ltd. (Shanghai, China), and processed using the Illumina MiSeq platform (Illumina Inc., San Diego, CA, USA).

### 2.3. DSV Culture and Detection

DSV was brought from China General Microbiological Culture Collection Center (CGMCC, Beijing, China) and resuscitated using their protocol. The DSV were then cultivated in a specific culture medium and placed in an anaerobic workstation for a week. Then, the cultures were centrifuged at 3000 rpm for 5 min at 4 °C, washed twice with sterile anaerobic phosphate-buffered saline (PBS), and resuspended to achieve a final concentration of 1.4 × 10^9^ CFU/mL under strictly anaerobic conditions. Then, polymerase chain reaction (PCR) was used to determine whether there were any other bacterial breeds in the solution. To detect DSV colonization in mice colons, DNA was extracted from the mice stool samples using a QIAamp DNA Stool Mini Kit (Qiagen, Hilden, Germany). Primers for DSV genes were used to detect faecal microbiota using real-time PCR (Appendix A). DNA sampling was performed in triplicate, and the comparative CT was used to determine the expression of specific genes of targeted bacterial genera.

### 2.4. 16S rRNA Sequencing and Sequence Processing

The V3–V4 region of the 16S ribosomal RNA (rRNA) gene from human and rat samples was amplified. Sequencing libraries were generated using the Ion Plus Fragment Library Kit 48rxns (Thermo Fisher Scientific, Waltham, MA, USA) following the manufacturer’s recommendations. The library was sequenced on the Ion S5 XL platform.

### 2.5. Detection of Serum Lipid and Liver Enzymes

Murine blood was incubated at 37 °C for 30 min, and the serum was obtained by centrifugation. Total cholesterol (TC), LDL-c, and high-density lipoprotein cholesterol (HDL-c) were detected after 16 h of fasting. Biochemical parameters, including serum alanine aminotransferase (ALT), aspartate aminotransferase (AST), and lactate dehydrogenase (LDH), were determined using an automatic biochemical analyser (HITACHI, Tokyo, Japan).

### 2.6. RNA Isolation and Real-Time Quantitative PCR

Total RNA was extracted from murine proximal colon tissue and liver samples using TRIzol reagent (TaKaRa, Shiga, Japan), and was reverse transcribed into cDNA using PrimeScript RT Master Mix (TaKaRa) according to the manufacturer’s instructions.

Real-time PCR was performed using the SYBR Green Master Mix (Yeasen, Shanghai, China) and the 7500 instrument (Applied Biosystems, Waltham, MA, USA). Three biological replicates were completed for all samples. The primer sequences are listed in Appendix A. The expression relative to GAPDH was determined using the ΔΔCt method.

### 2.7. Hematoxylin and Eosin Staining (HE)

Murine colon and liver tissues were incubated in 10% formaldehyde for 48 h, stored in ethanol, and embedded in paraffin. The tissue blocks were cut into 4-μm sections and stained with HE (BASO, Zhuhai, China). The slides were observed under a microscope (Olympus, Tokyo, Japan). Pathologists blinded to the treatment evaluated the slides. HE for pathological analysis was based on a previous protocol [35].

### 2.8. Immunohistochemistry (IHC)

Rat colonic sections were fixed in 4% formalin, embedded in paraffin, dewaxed using xylene, and then rehydrated using a graded series of alcohols. Subsequently, heat-induced epitope recovery was performed on the slides using the pressure cooker antigen repair method in Ethylene Diamine Tetraacetic Acid (EDTA), and endoperoxidase activity was blocked using 3% H_2_O_2_ for 30 min. The slices were incubated with zonula occludens-1 (ZO-1) (1:100; Thermo Fisher Scientific) and occludin (OCLN) (1:100; Thermo Fisher Scientific) antibodies overnight at 4 °C. Later, the slides used for IHC staining were incubated with horseradish peroxidase (HRP)-conjugated secondary antibodies (anti-rabbit IgG polymer) for 50 min, followed by DAB and haematoxylin staining. Finally, the results of IHC staining were observed and photographed using a microscope (Zeiss, oberkochen, German). The staining intensity of the colonic tissues was evaluated using image analysis software (Nikon, Tokyo, Japan).

### 2.9. Oil Red O (ORO) Staining

Rat liver tissue frozen slices were reheated, dried, fixed in 4% paraformaldehyde for 15 min, washed with tap water, and then dried again. To detect neutral lipids and lipid droplets, liver tissues were stained with ORO solution (Servicebio, Wuhan, China) for 8–10 min in the dark and covered with a lid while dyeing. The tissues were observed under a fluorescence microscope (Nikon), and the ORO staining intensity was analysed using NIS-Elements (v5.11; Nikon).

### 2.10. Analysis of Serum LPS and H_2_S Levels

Murine serum LPS levels were measured using an LPS Elisa kit (Jianglai Biotechnology, Shanghai, China), and the serum H_2_S levels of mice were measured using a mouse H_2_S Elisa kit (Solarbio, Beijing, China). The preparation of all reagents, working standards, and protocols was in accordance with the manufacturer’s instructions. Absorbance was read at 450 nm using a microplate spectrophotometer (Molecular Devices LLC, San Jose, CA, USA).

### 2.11. Statistical Analysis

Taxonomic assignments of the operational taxonomic units (OTUs) were performed using an RDP clarifier in accordance with the Greengenes database, based on their similarity with each other according to a specific threshold (typically a sequence similarity of at least 97%). Several bioinformatics pipelines were implemented to analyse the 16S sequencing results, such as QIIME (Quantitative Insights Into Microbial Ecology; http://qiime.org/) accessed on 18 December 2019 and 4 March 2021, and MOTHUR (https://mothur.org/) accessed on 18 December 2019 and 4 March 2021, which were specifically designed for examining microbial communities.

All data were analysed using GraphPad Prism (version 7.0; GraphPad Software, Inc., San Diego, CA, USA). The significance of differences between groups was determined using Student’s *t*-test. The results are presented as means ± standard error of the mean (SEM), and *p* < 0.05 was considered statistically significant.

## 3. Results

### 3.1. HFD Contributed to Weight Gain and Increased Lipid Levels in Rats

Six-week-old rats were fed the HFD for eight weeks, which significantly promoted weight gain (Figure 1A). We also measured serum lipid levels and found higher TC and LDL-c levels in the HFD group compared to the control and HFD+antibiotics groups (Figure 1B). In contrast, HDL-c was lower in the HFD group compared to the control group, while there was no significant difference between the HFD and HFD+antibiotics groups (Figure 1B). Similar results were observed for liver fat vesicles, although the vesicles were reduced by antibiotic administration (Figure 1C).

### 3.2. HFD Enhanced Mucosal Inflammation and Induced Mucosal Barrier Dysfunction in the Colon

The effect of HFD on colon morphology was also investigated. The HFD group had the shortest colons among all groups, but this trend was attenuated by antibiotics (Figure 2A,B). To investigate the effect of HFD on mucosal inflammation, HE staining of colonic tissues and gene expression analysis of inflammatory cytokines were performed. HE staining showed that the HFD and HFD+antibiotics groups had more inflammatory cell infiltration in the colon epithelium compared to the control group (Figure 2C). The pathological scores confirmed this trend (Figure 2D). In accordance with the HE staining results, tumour necrosis factor (TNF)-α and interleukin (IL)-β mRNA expression levels were higher in the HFD and HFD+antibiotics groups compared to the control group. Surprisingly, inflammatory genes exhibited the highest expression in rats fed the HFD and antibiotics (Figure 2E).

Next, we assessed ZO-1 and OCLN, which are important tight junction proteins, to evaluate the effect of HFD on barrier function. The HFD group had lower expressions of ZO-1 and OCLN than the control and HFD+antibiotics groups, while the combination of HFD and antibiotics also decreased ZO-1 and OCLN compared to the HFD group (Figure 2F). IHC staining was also used to assess the barrier function of the colon in different groups. HFD depleted ZO-1 and OCLN at epithelial junctions, although this effect was attenuated by the antibiotic cocktail in rats (Figure 2G,H). Our results showed that an HFD can trigger an inflammatory response in the colon epithelium. Furthermore, in agreement with previous studies, the antibiotic cocktail could enhance the inflammatory storm. Meanwhile, the barrier function of the colon was impaired by the HFD.

### 3.3. HFD Promoted Liver Inflammation and Pre-Metastatic Niche Formation

We then investigated whether the HFD had an effect on liver pre-metastatic niche formation. HFD elevated LPS levels, but this effect was suppressed when the HFD and antibiotics were administered simultaneously (Figure 3A). Rat serum was collected to detect transaminase levels after HFD administration. The HFD elevated ALT, AST, and LDH levels compared to the control group, but the antibiotics cocktail reduced their levels (Figure 3B). HE staining demonstrated that inflammatory cells were enriched in the HFD group (Figure 3C). These findings were confirmed by quantitative PCR. The expression of Toll-like receptor 4 (TLR4), which is activated by LPS and can trigger the production of proinflammatory cytokines, was higher in the HFD+antibiotics group, while the HFD group also showed elevated expression (Figure 3D). A similar trend was observed in inflammatory genes, such as IL-6, TNF-α, and IL-β mRNA (Figure 3D). Matrix metallopeptidase 2 (MMP2), matrix metallopeptidase 9 (MMP9), C-X-C chemokine ligand 12 (CXCL12), and fibronectin are molecular markers of pre-metastatic niche formation. All of these markers were significantly elevated after HFD but decreased when the rats consumed the antibiotic cocktail (Figure 3E). These findings suggest that HFD damages liver tissues and promotes the formation of pre-metastatic niches, but these effects are attenuated by antibiotics.

### 3.4. HFD Modulated the Gut Microbiome

Illumina MiSeq sequencing was used to determine whether HFD changed the gut microbiota. The majority of phyla identified in the colon were *Bacteroidetes*, *Firmicutes*, *Actinobacteria*, and *Proteobacteria* (Figure 4A). *Proteobacteria* (Gram-negative) and *Bacteroidetes* (Gram-negative) were more numerous in the HFD group compared to the control group (Figure 4B,C). Meanwhile, at the genus level, *Helicobacter*, *Ruminococcus*, *Desulfibrionales*, *Clostridium*, *XlVb*, *Oscillibacter aunclassified Firmicuts*, and *Olsenella* were significantly increased in the HFD group compared to the control group (Figure 4D,E).

### 3.5. DSV Was Enriched in CRC Liver Metastasis Patients

To clarify the mechanism underlying the association between gut microbiota and CRC liver metastasis, we used the Illumina MiSeq sequencing platform to analyse the gut microbiota in CRC patients with and without liver metastases. At the phylum level, *Bacteroides* and *Proteobacteria* were enriched in CRC patients without liver metastases (Figure 4A). A similar trend was found in the genus *Clostridium*, *Faecalibacterium* and *Prevotella* (Figure 4B). Meanwhile, *DSV*, *Barnesiella*, *Gemmiger*, and *Ruminococcus* were enriched in the metastasis group (Figure 4C). In particular, DSV, a Gram-negative bacterial species from the SRB group, was significantly increased in the metastatic group (Figure 4C). FARROTAX also confirmed that the sulphide reduction capacity was much greater in the metastatic group (Figure 4D). In addition, the evolutionary trees of species suggested that all the bacteria that varied between the two groups were Gram-negative (Figure 4E). Surprisingly, compared to a normal diet, rats fed an HFD for eight weeks had a higher relative abundance of DSV (Figure 4F,G and Figure 5).

### 3.6. DSV Caused Colon Damage and Promoted Liver Pre-Metastatic Niche Formation

To determine the effect of DSV on colon mucosa and pre-metastatic niche formation, faeces were collected, and qPCR was performed to assess DSV abundance in the colon. The mice pre-treated by oral gavage of the antibiotic cocktail for 2 weeks before DSV gavage had a higher relative abundance of DSV compared to the DSV and control groups (Figure 6A). Furthermore, the trend for H_2_S, a metabolic product of DSV in the gut, was similar to that for DSV abundance (Figure 6B). LPS, derived from Gram-negative bacteria, was higher in the DSV group, but much lower compared to the DSV+antibiotics group (Figure 6C).

HE staining showed greater inflammatory cell accumulation in the DSV than the control group, whereas the DSV+antibiotics group had the most infiltration (as confirmed by the pathological scores; (Figure 6D,E). DSV gavage also increased TNF-α and IL-1β mRNA expression, which indicated that DSV induced and enhanced the inflammatory response in colon mucosa (Figure 6F). Moreover, there was an obvious reduction in the expression of the barrier-related proteins, ZO-1 and OCLN, in the DSV and DSV+antibiotics groups compared to the control group, but there was no difference between the two DSV groups (Figure 6G). These findings indicated that DSV breeding may impair intestinal barrier function.

We next evaluated the effect of DSV on the liver. Serum ALT and AST were significantly elevated by DSV, reflecting a critical role in the gut microbiota (Figure 6H). In terms of inflammation, DSV gavage alone for eight weeks failed to augment the inflammatory response in the liver, but that situation changed as more DSV bred in the gut. TLR4, which is activated by LPS and leads to the production of proinflammatory cytokines, was over-expressed along with other inflammatory cytokines, such as IL-6, TNF-α, and IL-β, in the DSV+antibiotics group (Figure 6I). In addition, MMP2, MMP9, and CXCL12, which are associated with liver pre-metastatic niche formation, were also significantly increased as more DSV bred in the gut (Figure 6J). These findings highlight that DSV has a great influence on gut mucosa and liver niche formation.

## 4. Discussion

In this study, HFD induced intestinal microbiota alterations, especially an increase in the Gram-negative microbiota, similar to the changes seen in faecal samples of CRC liver metastasis patients. Changes in the intestinal microenvironment, including inflammation and barrier dysfunction, can promote CRC initiation [11,36]. Studies have shown that HFD exacerbates intestinal inflammation and barrier dysfunction [37]. In this study, following administration of an HFD for eight weeks, severe intestinal inflammation occurred in rats. The intestinal barrier markers, including ZO-1 and OCLN, were decreased in HFD-fed rats, and antibiotics alleviated the barrier dysfunction. However, the intestinal inflammation was most severe in the mice fed the HFD and antibiotics. A similar phenomenon was previously reported, i.e., an HFD and antibiotics had a synergistic effect on intestinal inflammation due to reduced mitochondrial bioenergetics [38].

There is evidence that LPS may be involved in the inflammation induced by HFD [39,40]. We observed increased serum LPS in the HFD mice, likely induced by Gram-negative microbiota. LPS can bind with TLR4 and stimulate liver inflammation, which induces CRC liver metastasis [21,41]. HFD exacerbated liver inflammation and increased the cytokines associated with tumour metastasis, possibly due to high serum LPS levels. In addition, lipid deposition was found in HFD mice livers; this is a known risk factor for non-alcoholic fatty liver disease and hepatocellular carcinoma [42,43]. Gradually, through lipid deposition, pro-metastatic cytokine production, and inflammation, a microenvironment conducive to metastasis was established.

Changes in gut microbiota, including *Enterococcus faecalis*, *Escherichia coli*, *Bacteroides fragilis*, and *Fusobacterium nucleatum*, in CRC, have been reported previously [15,27]. To investigate how microbiota alterations influence the microenvironment in the colorectum and liver, we performed 16S rRNA sequencing of the faecal samples from CRC liver metastasis patients and found an abundance of Gram-negative bacteria. Furthermore, our data demonstrated, for the first time, DSV enrichment in the metastasis compared to the non-metastasis group. It has been suggested that the abundance of *sulphidogenic bacteria* may be related to a diet high in animal protein and fat [30]. Meanwhile, enrichment was also found in faecal samples from HFD-fed rats. As an SRB, DSV produces H_2_S, which may contribute to colitis and CRC [29,44,45]. DSV was reported to be abundant in fatty liver models [46,47]. Therefore, we speculated that DSV might be responsible for the carcinogenesis and liver metastasis seen in CRC.

Antibiotic gavage before DSV depleted the intestinal microbiota and made DSV the dominant bacteria in the gut [33]. After DSV gavage, mice showed high serum H_2_S and LPS levels, as well as severe intestinal inflammation and barrier dysfunction, which could have been induced by LPS [48]. H_2_S also damages the intestinal barrier by reducing disulphide bonds [49,50]. Liver inflammation and pro-metastatic cytokine levels increased with increasing DSV abundance. Pre-metastatic niches are microenvironments that make distant organs conducive to tumour metastasis [51]. An increase in MMP2, MMP9, and CXCL12 is considered an element of extracellular matrix remodelling, which is critical for the formation of pre-metastatic niches [23,52,53]. Therefore, DSV may enhance the liver metastasis potential of CRC through extracellular matrix remodelling and pre-metastatic niche formation.

Our study demonstrated HFD-induced changes in colorectal and liver microenvironments. Furthermore, we first demonstrated DSV enrichment in the faecal samples of CRC liver metastasis patients and investigated the DSV-induced pro-metastatic microenvironment in the liver. This study could pave the way for investigations of HFD-induced gut microbiota disorders, especially DSV abundance, in CRC liver metastasis. It provided a reference for the evaluation of liver metastasis potential in CRC. However, the research was limited by the lack of exploration of the underlying mechanisms; further research is required to determine the mechanisms by which HFD and DSV promote carcinogenesis and metastasis in CRC.

## 5. Conclusions

HFD can induce intestinal microbiota alterations, reflected in an increase in the Gram-negative microbiota. Furthermore, HFD can induce colonic inflammation, barrier dysfunction, and liver microenvironment remodelling, by changing the intestinal microbiota and promoting CRC initiation and liver metastasis. DSV could also play a critical role in CRC initiation and the promotion of a pre-metastatic microenvironment in the liver.

## Figures and Tables

**Figure 1 cancers-14-02573-f001:**
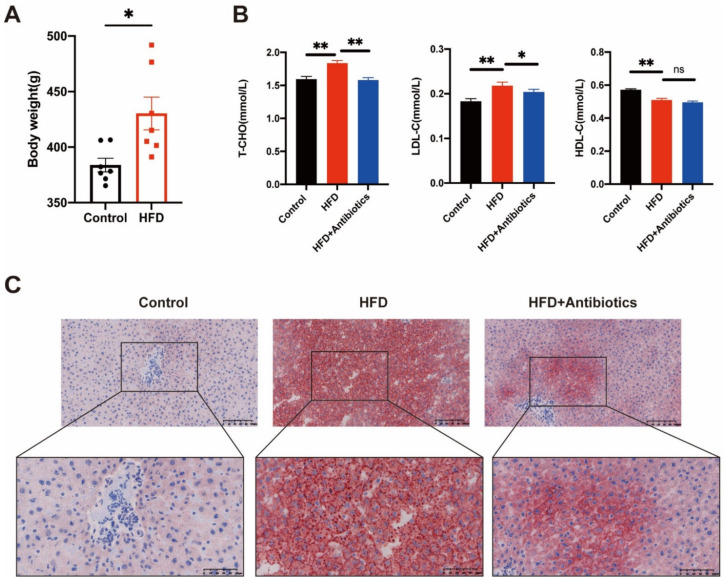
Treatment of a high-fat diet could induce lipid accumulation and dyslipidemia in rats. (**A**) The administration of high-fat food for eight weeks effectively increased body weight gain in Wistar rats. (**B**) Levels of HDL-C, T-CHO, and LDL-C in rats at eight weeks of administration. (**C**) Hepatic oil red O stain of liver tissue sections collected from rats. Scale bars respectively represent 100 μm and 50 μm. Data represent means ± SEM. * *p* < 0.05; ** *p* < 0.01, ns: no significant. (Control, n = 7; HFD, n = 7; HFD+Antibiotics, n = 8).

**Figure 2 cancers-14-02573-f002:**
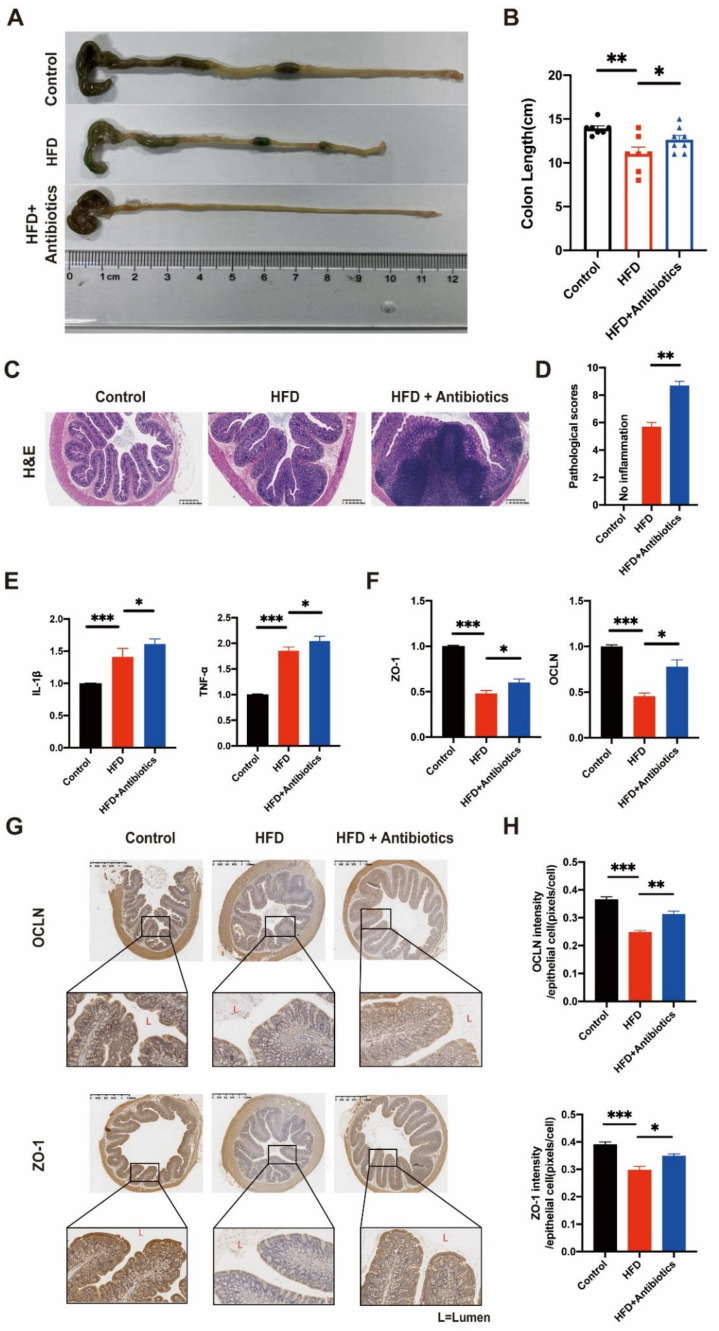
Treatment of a high-fat diet could induce intestinal inflammation and barrier dysfunction in rats. (**A**,**B**) Rat colons were removed and shown in (**A**), colon length from the Control, HFD, and HFD+Antibiotics groups were measured. (**C**,**D**) Haematoxylin and eosin staining of colon tissue sections collected from rats treated as above. Scale bars represent 400 μm. HE was calculated histopathological inflammation scores (**D**). (**E**,**F**) mRNA expression of ZO-1, OCLN, TNF-α, and IL-1β are shown as relative fold to control normalized to GAPDH. (**G**,**H**) Immunohistochemical staining of ZO-1 and OCLN was detected in three groups (**G**), and quantitative evaluation (**H**) of the expression of ZO-1 and OCLN in the intestinal mucosa. Scale bars represent 1.25 mm. Data represent means ± SEM. * *p* < 0.05; ** *p* < 0.01; *** *p* < 0.001. (Control, n = 7; HFD, n = 7; HFD+Antibitics, n = 8).

**Figure 3 cancers-14-02573-f003:**
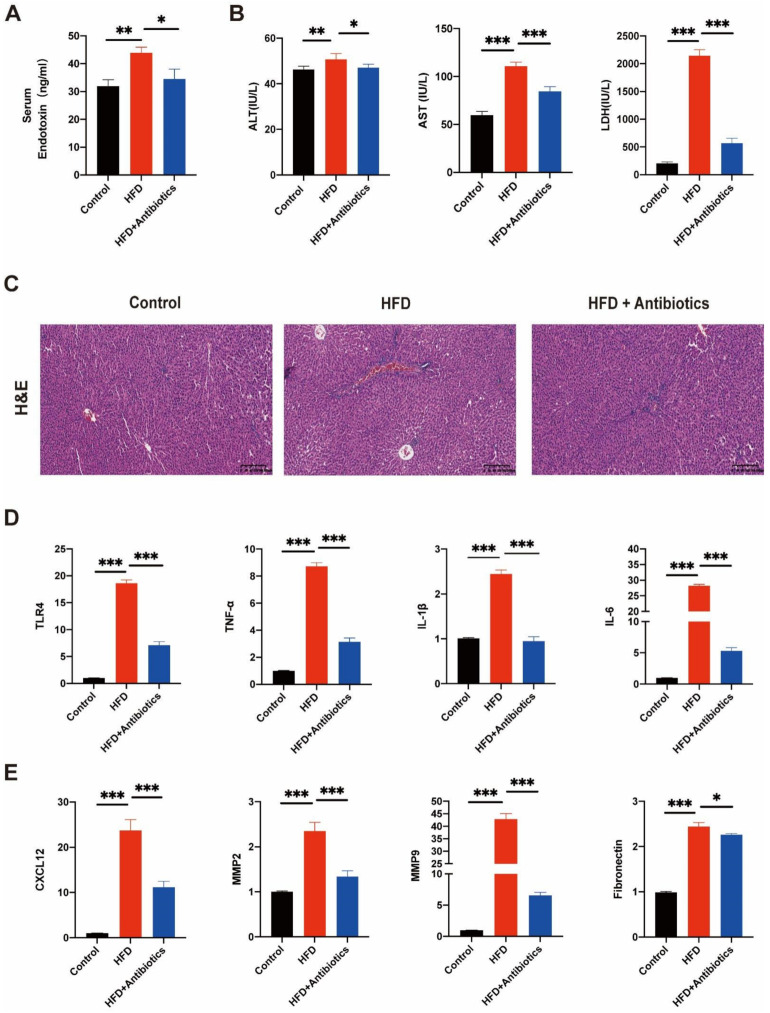
Treatment of a high-fat diet could induce liver damage and be engaged in the construction of metastasis microenvironment in rats. (**A**) The level of serum endotoxin was evaluated by serum endotoxin production assay. (**B**) Levels of serum ALT, AST, and LDH in rats at eight weeks of administration. (**C**) Haematoxylin and eosin staining of liver tissue sections collected from rats treated as above. Scale bars represent 200 μm. (**D**,**E**) mRNA expression of IL-1β, IL-6, TNF-α, TLR4, MMP9, Fibronectin, MMP2, and CXCL12 are shown as relative fold to control normalized to GAPDH. Data represent means ± SEM. * *p* < 0.05; ** *p* < 0.01; *** *p* < 0.001. (Control, n = 7; HFD, n = 7; HFD+Antibitics, n = 8).

**Figure 4 cancers-14-02573-f004:**
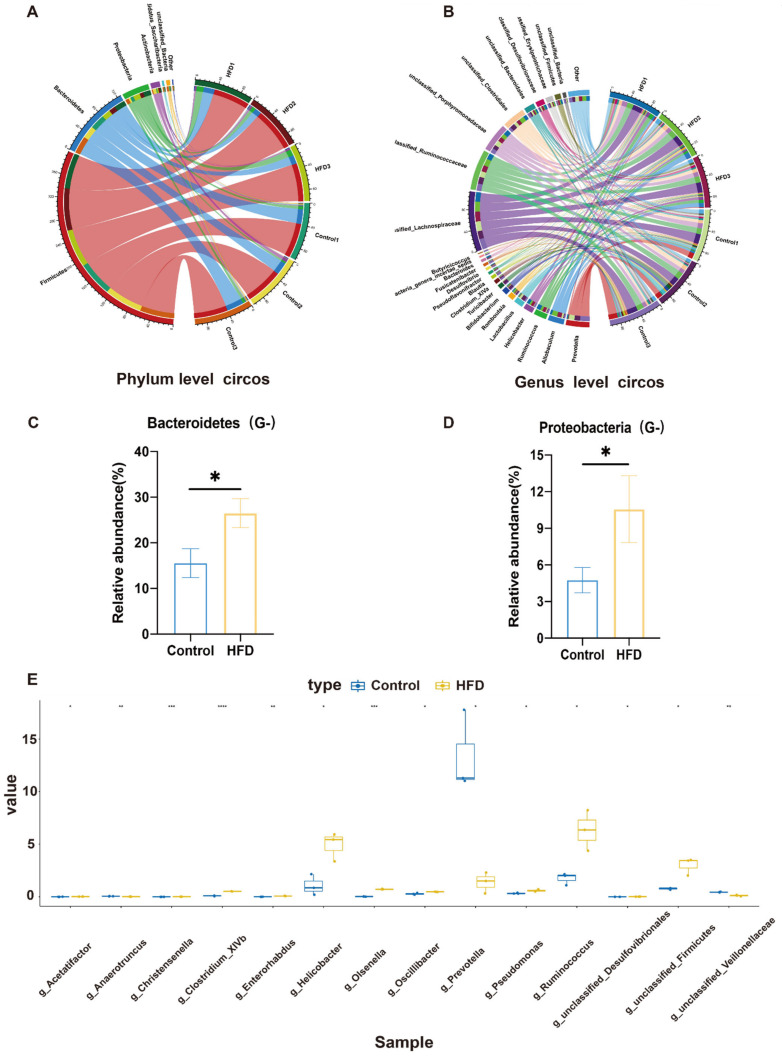
Overview of faecal flora distribution in rats after HFD treatment for eight weeks. (**A**) Phylum level circus of faecal bacteria. (**B**) Genus level circus of faecal bacteria. (**C**,**D**) Representative bacteria at the phylum level with differential expression. (**E**) Representative bacteria at the genus level with differential expression. * *p* < 0.05; ** *p* < 0.01; *** *p* < 0.001; **** *p* <0.0001. (Control, n = 7; HFD, n = 7).

**Figure 5 cancers-14-02573-f005:**
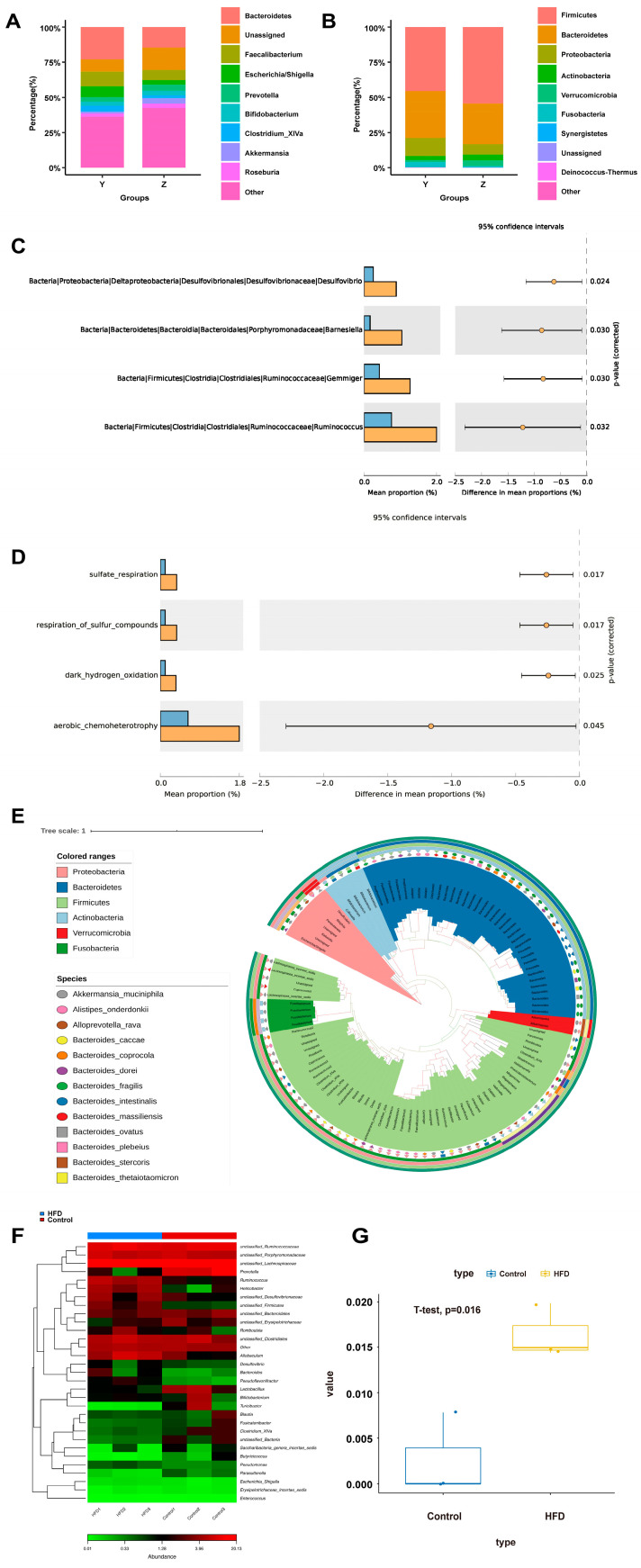
Faecal flora difference and function prediction between the human liver metastasis group and the non-metastatic group of colorectal cancer. (**A**) Faecal intestinal microbiota distribution at the genus level. (**B**) Faecal intestinal microbiota distribution at the phylum level. (**C**) Representative intestinal microbiota with differential expression between groups. (**D**) Function prediction of *Desulfovibrio*. (**E**) Evolutionary Trees of species reveals that all the bacteria varied in two groups were Gram-negative. (**F**,**G**) *Desulfovibrio* is highly expressed in HFD groups at the genus level. (Y: non-metastatic group, n = 10; Z: metastatic group, n = 13). * *p* < 0.05.

**Figure 6 cancers-14-02573-f006:**
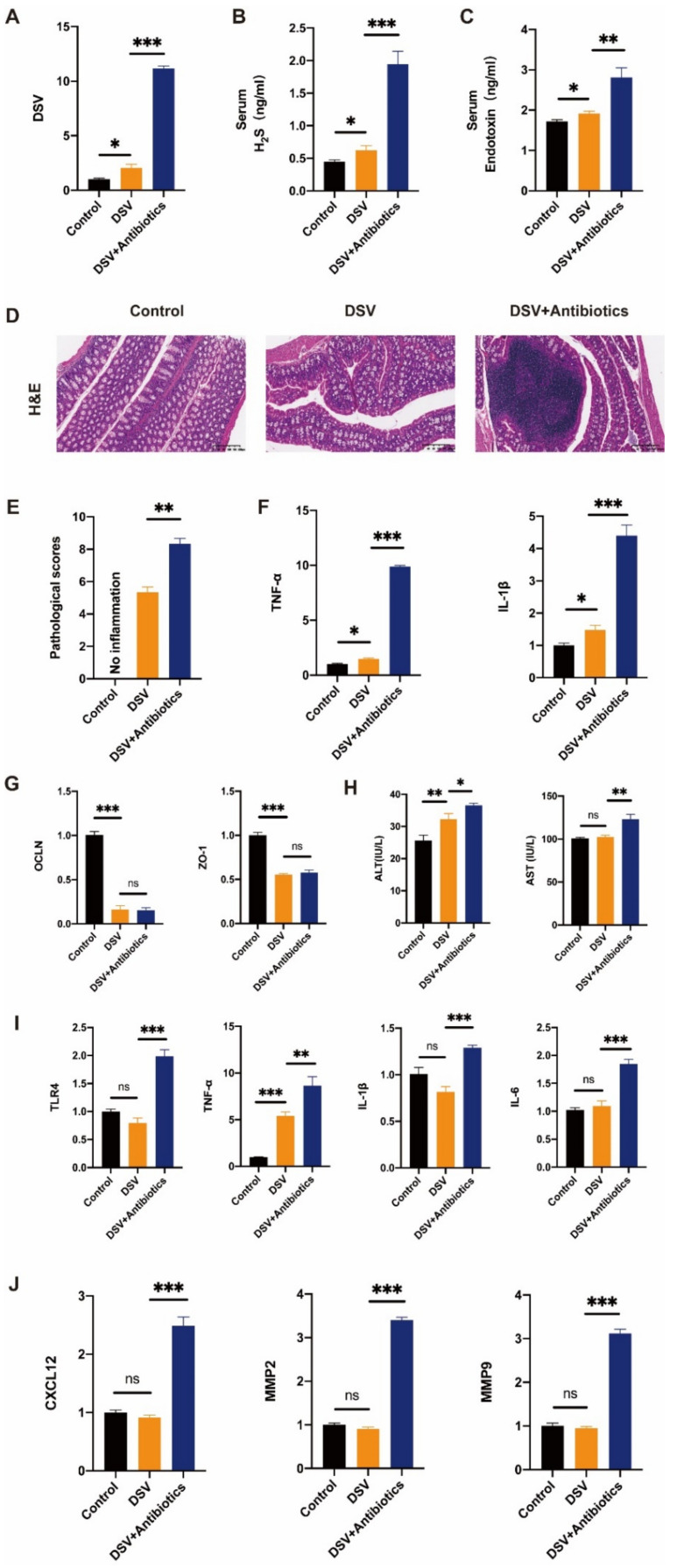
Oral gavage of DSV could exert an effect on the colon and liver and engage in the construction of metastasis microenvironment in mice. (**A**) The colonization of *Desulfovibrio* in the colon of different groups detected by qPCR. (**B**) The level of serum H_2_S in mice was evaluated by an H_2_S production assay. (**C**) The level of serum endotoxin in mice was evaluated by serum endotoxin production assay. (**D**) Haematoxylin and eosin staining of colon tissue sections collected from mice treated as above. (**E**) HE is calculated histopathological scores. (**F**,**G**) mRNA expression of IL-1β, TNF-α, ZO-1, and OCLN in intestinal mucosa are shown as relative fold to control normalized to GAPDH. (**H**) Levels of serum ALT and AST in mice at eight weeks of administration. (**I**) mRNA expression of TLR4, TNF-α, IL-1β, IL-6 are shown as relative fold to control normalized to GAPDH. (**J**) mRNA expression of MMP9, MMP2, and CXCL12 are shown as relative fold to control normalized to GAPDH. Data represent means ± SEM. * *p* < 0.05; ** *p* < 0.01; *** *p* < 0.001, ns: no significant. (Control, n = 7; DSV, n = 7; DSV+Antibiotics, n = 8).

## Data Availability

Publicly available datasets were analyzed in this study. This data can be found here: [https://ngdc.cncb.ac.cn (accessed on 12 March 2020)/HRA001836].

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
