# Peer review of "High-Fat Diet Enhances the Liver Metastasis Potential of Colorectal Cancer through Microbiota Dysbiosis"

_cancers, 2022, doi:10.3390/cancers14112573_

Round 1

Reviewer 1 Report

The manuscript „High-Fat Diet enhances the liver metastasis potential of CRC through microbiota dysbiosis“ aims to investigate the correlation between HFD-induced intestinal dysbiosis, microbiota composition, colorectal cancer, and liver metastasis potential. Moreover, the influence of Desulfovibrio on carcinogenesis and metastasis has been evaluated. 

Overall, the manuscript is well organized, the references are adequate, and the results are interpreted by an appropriate number of pictures and graphs&pictures of high quality.

I have several comments and suggestions:

  • I suggest using colorectal cancer instead of the abbreviation CRC in the Title of the manuscript
  • Italic style should be used to term bacterial taxa and kept throughout the whole manuscript (Bacteroides fragilisEscherichia coli, etc...)
  • I would suggest changing the term intestinal flora to intestinal/gut microbiota
  • Correct the formatting error in Section Title 2.4 16S rRNA sequencing
  • Please check the correctness of the sequencing method for the determination of bacterial composition – there is an Ion S5 XL platform in the methodology, but the authors have also stated that the Illumina platform was used for 16 S rRNA sequencing

Reviewer 2 Report

The article entitled "High-Fat Diet enhances the liver metastasis potential of CRC through microbiota dysbiosis " presents some potentially interesting findings. Nevertheless, some questions should be addressed in order to improve the scientific quality:

  • The article will benefit from English language proofreading. The grammar problems make it difficult to follow.
  • In the Methods section "2.8. Immunohistochemistry", the fixation method and the antigen retrieval method should be stated, for the sake of reproducibility. Fixative solution should also be mentioned in section "2.9. Oil Red O (ORO) Staining".
  • The titles of the subsections in the Results are ambiguous. Each title should state the results obtained (similarly to the Figure legends), not ambiguous sentences such as "influence the metabolic state of rats" or "change mucosal barrier function".
  • Figure 4A is too small and all Figure 4 is blurry. Same for Figure 5C, 5D and 5F.

Round 2

Reviewer 2 Report

Thank you very much for your effort. All my comments have been addressed, except for the antigen retrieval solution used in "2.8. Immunohistochemistry (IHC)", could you please mention what was it?
Thank you very much.

Author Response

Thanks for your comments, we are sorry we did not give a reply on the matter of antigen retrieval solution in the new manuscript version. We used the Ethylene Diamine Tetraacetic Acid as antigen retrieval solution.